# Development and Validation of a Novel Tool for Assessing the Environmental Impact of 3D Printing Technologies: A Pharmaceutical Perspective

**DOI:** 10.3390/pharmaceutics14050933

**Published:** 2022-04-25

**Authors:** Souha H. Youssef, Sadikalmahdi Abdella, Sanjay Garg

**Affiliations:** 1Pharmaceutical Innovation and Development Group (PIDG), Clinical and Health Sciences, University of South Australia, Adelaide, SA 5000, Australia; souha.youssef@mymail.unisa.edu.au (S.H.Y.); sadikalmahdi.abdella@mymail.unisa.edu.au (S.A.); 2College of Health Sciences, Addis Ababa University, Zambia Street, Addis Ababa P.O. Box 9086, Ethiopia

**Keywords:** greenness, 3D printing, environmental impact, green chemistry, tool

## Abstract

Technological advancements have created infinite opportunities and rendered our life easier at several fronts. Nonetheless, the environment has suffered the aftermaths of modernization. Ironically, the pharmaceutical industry was found to be a significant contributor to environmental deterioration. To tackle this issue, continuous eco-evaluation of newly introduced technologies is crucial. Three-dimensional printing (3DP) is rapidly establishing its routes in different industries. Interestingly, 3DP is revolutionising the production of pharmaceuticals and is regarded as a promising approach for the fabrication of patient-centric formulations. Despite the increasing applications in the pharmaceutical field, tools that evaluate the environmental impacts of 3DP are lacking. Energy and solvent consumption, waste generation, and disposal are the main associated factors that present major concerns. For the first time, we are proposing a quantitative tool, the index of Greenness Assessment of Printed Pharmaceuticals (iGAPP), that evaluates the greenness of the different 3DP technologies used in the pharmaceutical industry. The tool provides a colour-coded pictogram and a numerical score indicating the overall greenness of the employed printing method. Validation was performed by constructing the greenness profile of selected formulations produced using the different 3DP techniques. This tool is simple to use and indicates the greenness level of the procedures involved, thereby creating an opportunity to modify the processes for more sustainable practices.

## 1. Introduction

### 1.1. History of Environmental Awareness

Although the technological advancements in modern times have improved the quality of life, facilitated our daily activities, and opened the doors to endless possibilities, the environment has suffered the repercussions in different ways such as pollution, global warming, and ecological issues [1]. In 1962, Rachel Carson was the first to shed light on the devastating effects of some chemicals on ecosystems in her book “Silent Spring”. The book was an eye-opener to both scientists and the public and acted as a foundation stone for many movements towards protecting the environment. Soon after, the US Environmental Protection Agency (EPA) was established in 1970 and passed several legislations to ensure human safety and environmental sustainability [2]. Subsequently, many other countries including Canada (Environment Canada), Australia (National Environment Protection Council), and England (Environment Agency) established their regulatory bodies for environmental affairs. The early efforts of EPA were dedicated to restoring polluted areas and banning toxins with an obvious threat to humans and ecosystems. However, as more awareness was raised, scientists started to contemplate pollution prevention as a feasible approach as opposed to focusing on clean-up measures. The notion was met with international acceptance and the Organization for Economic Co-operation and Development, with 38 member countries, joined efforts and released recommendations for pollution prevention.

The term green chemistry (GC) was coined by Paul Anastas in 1991 and approved as a legitimate field of scientific research [3]. Later, Anastas and Warner co-authored their revolutionary book, “Green Chemistry: Theory and Practice”, which serves as the basis for GC practices to date. The book discusses 12 principles to implement more eco-friendly procedures [4]. These principles were recently summarized by the acronym “PRODUCTIVELY” (Table 1) [5].

The development of novel pharmaceutical products based on organic synthesis marked a turning point in medical care, improving treatment efficiency and reducing deaths and hospitalization. Nonetheless, the high demand for these products placed the pharmaceutical industry among the major contributors to the generation of chemical waste and pollutants [6]. In fact, studies showed that the pharmaceutical industry generated more byproducts than fine chemicals, bulk chemicals, and oil refining sectors [7] and contributed remarkably towards the emission of greenhouse gases [8]. The industry soon realized the potential impact on the environment and took steps towards greener practices. For instance, Patheon, Biogen, Johnson & Johnson, Genentech, and Novartis approved the American Business Act on Climate Pledge that aimed to reduce their greenhouse gas emissions, water use, and waste besides favouring the use of renewable energy [9]. Some pharmaceutical companies have released solvent selection guides demonstrating the greenness of different solvents and recommended substitutions of hazardous solvents [10]. Unsurprisingly, the American Chemistry Society Green Chemistry Institute (ACS GCI) established an industrial roundtable for the pharmaceutical industry, in order to improve the sustainability of pharmaceutical development processes [11]. The field of GC marked the cornerstone to environmental awareness and paved the way for incorporating environmental policies in all sectors of production to design products and processes without compromising human health and the environment.

### 1.2. Greenness Assessment Tools

Continuous efforts have been made at the industrial, academic, and research institutions levels to ensure that analytical, production, and formulation development processes impart minimal human and environmental harm. Among these, the adoption of the sustainability concept by most industries and the development of greenness-assessment tools such as Life Cycle Assessment (LCA) and Environmental Impact Assessment (EIA) were the prominent ones [12]. The tools qualitatively describe the impact of products’ manufacturing process on the environment, giving an ecological indicator for the different stages of a product. Whilst the tools served as a benchmark and give general information about the impact of the processes on the environment, the methods do not allow objective comparisons between different processes, and some aspects were not clearly presented [13]. Several efforts such as converting general assessment terms such as “good”, “moderate” and “bad” into numerical scores were performed to improve LCA and EIA assessments tools but were found to be time-consuming [14,15]. After the emergence of green analytical chemistry (GAC) [16], scientists worked toward developing greener approaches for analysis in light of these concepts [17,18,19]. However, the evaluation of the greenness of a procedure was merely descriptive, making the overall assessment of the procedure challenging because analytical procedures require several steps. Later, tools that give a general indication of the greenness profile such as the national environmental methods index (NEMI) [20] and Raynie and Driver [21] followed by quantitative tools based on numerical scores such as analytical method volume intensity [22] and eco-scale assessment [23] were introduced. Finally, more inclusive tools, namely green analytical procedure index (GAPI) [24] and Analytical GREEnness Metric Approach and Software (AGREE), were then created by merging the benefits of quantitative scoring and visual illustration [25]. Figure 1 shows the greenness assessment of a high-performance liquid chromatographic method for the analysis of a binary mixture using different tools [17].

A study comparing the different assessment tools for analytical methods proved that merging visual representation with numerical scoring provided a complete evaluation of the greenness of the method in contrast to descriptive approaches [26].

Similar to analytical and synthesis procedures, drug products are performed over several steps that involve chemicals, energy consumption, and waste generation. Formulation scientists continuously explore different drug delivery systems and production technologies to increase the effectiveness and safety of drugs. Three-dimensional printing (3DP), also known as additive manufacturing or rapid prototyping, is among the innovative and promising technology introduced into the pharmaceutical sector recently. Despite few attempts to evaluate the environmental impact of 3DP technologies, there has been no report of tools used to evaluate the “greenness” of 3DP technologies used to produce drug dosage forms. Furthermore, previous studies were descriptive and limited to only three of the commonly used technologies in the pharmaceutical field [27]. Thus, there is a need for the development of an environmental assessment tool specific for 3DP of pharmaceuticals. The assessment of the newly adopted technology will reveal the environmental impacts of the technology on the environment and humans. In addition, it can aid in the implementation and sustainability of the technologies in drug production.

### 1.3. 3DP of Pharmaceuticals

3DP is an innovative technology causing a transformative change in medicines manufacture and drawing substantial attention owing to its potential to fabricate bespoken personalised dosage forms which are otherwise impossible with conventional manufacturing [28,29]. A growing evidence base has shown that 3DP is a promising technology that can address unmet clinical needs associated with traditional manufacturing methods often based on a “one-size-fits-all” approach. It is worthy to note that it offers several advantages such as an increase in patient adherence, decrease in pill burden, customisation, and personalization of medicines with individually adjusted doses, on-demand manufacturing, and the ability to fabricate complex solid dosage forms with high accuracy and precision, thereby improving effectiveness and safety of drugs [30].

The application of 3DP in the pharmaceutical sector started relatively late, and the beginning was marked by the approval of the first 3D printed levetiracetam tablet (Spritam^®^, Blue Ash, OH, USA) by the U.S. Food and Drug Administration (FDA). Since then, a significant increase in the number of pharmaceutical research focusing on 3DP of dosage forms aiming to develop personalized formulation and optimisation has been reported [31]. For instance, the number of published articles related to 3D printed dosage forms has increased by greater than 400% (from 145 in 2016 to 592 in 2020) in the web of science (Figure 2).

Marketwise, 3DP technology has made significant progress in diverse fields. The market value was estimated to be $7.34 billion in 2017 and is expected to grow to a value of $23.9 billion in 2022, and $35.6 billion in 2024 [32]. On the drug front, Spritam^®^ (developed by Aprecia Pharmaceuticals, Blue Ash, OH, USA) is the first and only FDA-approved 3D-printed drug product. More recently, another 3D-printed tablet (T19, developed by Triastek, Nanjing, China) received Investigational New Drug approval from FDA, increasing excitement in the technology. Over the last few decades, 3DP technology has undergone rapid growth, implying that it is clearly a growing industry [33].

#### 3DP Process and Methods

Despite the diversity of 3DP techniques, almost all 3DP methods involve the same basic procedures. Design and optimization of the 3D object using computer-aided design software, exporting a 3D model to a machine-readable 3D file format, slicing, 3DP technique selection, selection and processing of raw material, printing, and post-processing (quality assurance) are the major steps required to produce 3D printed objects [34].

Inkjet-based, extrusion-based, and laser-based methods are the major 3DP methods employed in the pharmaceutical field. The methods, in turn, comprise different techniques among which binder jetting (BJ), fused deposition modeling (FDM), pressure-assisted microsyringes (PAM), stereolithography (SLA), and selective laser sintering (SLS) are the major ones. The 3DP technologies primarily differ in the various layers of material that are formed and assembled to produce the desired dosage form [35] (Figure 3).

##### Fused Deposition Modeling

FDM, which is also called fused filament fabrication, is one of the most widely employed 3DP techniques because of its availability and use on both small and large scales (Figure 4a). It is a low-cost manufacturing process and allows the production of highly complex drugs with difficult geometries and offers good mechanical strength, as well as options to modify the drug release profiles.

It uses thermoplastic polymer filament as feedstock and consists of (a) a high-temperature liquefier block with a nozzle to melt the filament at a particular temperature, (b) a pinch roller mechanism to feed the filament into the liquefier block, (c) a heating cartridge to generate temperature for filament melting, and (d) a gantry system to move the print head in the horizontal direction to deposit the melted material on a build surface [33,36]. Two methods—direct and indirect—are used to incorporate drug/s to the filament [37]. In the direct method, filaments are prepared by melting active pharmaceutical ingredients and pharmaceutical-grade polymers using hot-melt extrusion [38]. Although this method of drug incorporation gives flexibility in terms of drug loading percentage, it is not generally suitable for thermosensitive drugs because it usually requires high temperatures to melt the polymers [39]. The indirect/diffusion method involves soaking or immersing blank filament in a saturated solution of the desired drug for a certain period to ensure diffusion of the drug into the polymeric matrix. The soaking of the drug mainly depends on the swelling of polymers when immersed in the solvent. Compared to the direct method, this method is time consuming, allows minimal drug loading, and results in higher drug wastage. Furthermore, the solvents used may be toxic and need the extra step of drying before final printing [40,41,42,43].

##### Stereolithography

SLA creates 3D objects by selective photopolymerization of liquid photosensitive resins using ultraviolet laser sources [44,45]. Briefly, a thin layer of resin liquid composed of drug and photoinitiator is scanned point by point to polymerize and attach to the building platform. The platform moves up or down, depending on the approach used, to the extent which depends on the thickness of the layer. Subsequently, the liquid resin is redistributed above the previously formed layer, and the process repeats until the final object is formed (Figure 4b). Finally, the curing step which helps to remove abundant resin and photoinitiator as well as improve mechanical strength is performed [46].

##### Selective Laser Sintering

SLS, also known as the powder bed fusion technique, uses a CO_2_ laser beam to heat and fuse selected regions of powders in each layer with high precision (Figure 4c). It is a promising technology that offers a high resolution, single-step, and solvent-free method for drug delivery. SLS is composed of three main systems, namely powder bed, spreading platform, and laser. In short, the spreading system spreads the powder uniformly on the platform, and a rollerblade is used to even the surface. The powder is then heated to a temperature below its melting point using the laser. The powder should be slowly cooled after printing to avoid stress and curl distortions [34,47]. Additional layers are deposited and fused until the final product is formed [48].

##### Semisolid Extrusion (SSE)

SSE, also known as PAM extrusion technology, uses gel/paste as feedstock to produce the dosage forms [49]. The feedstock is extruded evenly through a syringe-based print head under pressure and deposits layer by layer on the printing platform according to the modeling software (Figure 4d). Compared with other printing technologies, SSE is suitable for producing thermosensitive drugs because the printing can be done at a lower temperature. SSE might require the use of organic solvents to prepare the paste, which may cause the problem of residual organic solvents in the dosage forms [50]. Furthermore, obtaining a suitable gel with the right viscosity and drying the final product have been some of the challenges associated with this printing technique.

**Figure 4 pharmaceutics-14-00933-f004:**
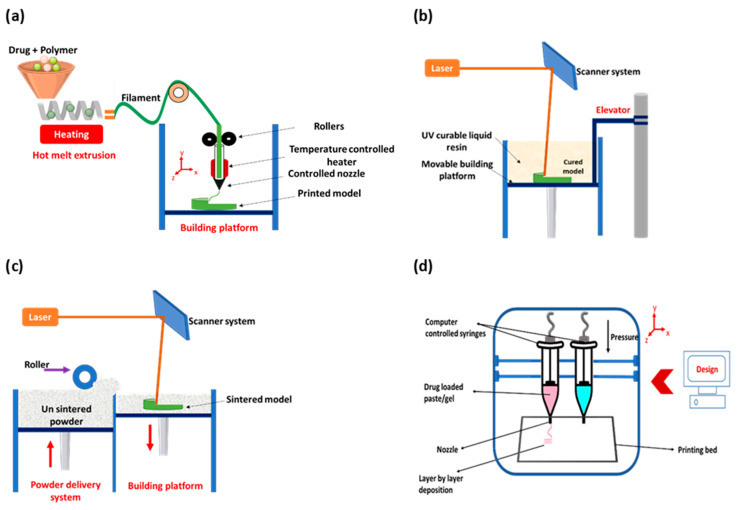
A schematic diagram illustrating (**a**) FDM (**b**) SLA (**c**) SLS (**d**) SSE printers (Adapted from [51]).

##### Binder Jetting (BJ)

BJ, also known as drop-on-powder, utilizes an ink-jet head that can jet-dispense a liquid binder solution onto a flattened powder bed. Briefly, a layer of powder is evenly spread on the build platform using a roller, and the print head ejects droplets containing active pharmaceutical ingredients or a binder onto the powder bed at an accurate speed according to the specified pattern designed in the computer. After printing one layer, the platform is lowered one layer along the vertical axis, and then a new layer of powder is spread over the previous layer. This process is repeated until the dosage forms are completed (Figure 5). The post-processing involves the elimination of residual solvent and the recovery of unprocessed powder, which supports the holistic structure. Because of the benefits of its pharmaceutical application, such as easy application and fixation, reduced development time, and favorable aesthetic results, BJ 3DP technology can be better adapted to the manufacturing of dosage forms. One of the main reasons for this is that the starting materials (such as powders and binder solutions) have already been widely used in the pharmaceutical industry. However, there are some associated challenges, such as the additional drying process to eradicate residual solvents and to improve the physical resistance of the 3D-printed constructs.

### 1.4. Environmental Impacts of 3DP

The application of 3DP in drug delivery is particularly promising due to the heralding move towards personalised medicine. However, with sincere emphasis on manufacturing methods with minimal harm to the environment, it is necessary to thoroughly evaluate newly introduced technologies. The environmental impact of 3DP is still indecisive. A review refers to 3DP as sustainable manufacturing, stating that expenditure of resources, material-related energy demands, and waste generation are reduced, as well as the possibility of recycling [52,53]—additionally, faster 3DP processes, lower energy consumption and carbon dioxide emission [54]. On the other hand, Shuaib et al. [13] discussed negative impacts of 3DP such as excessive energy consumption and waste, and freshwater and marine eutrophication. Some studies highlighted the release of volatile organic compounds during extrusion of some filaments for FDM printers which are carcinogenic; however, the maximum estimated levels were found not to pose a risk on human health [55].

LCA and design for environment are two methods used to assess the environmental impacts of production processes [56]. Moreover, other methods have been proposed to evaluate the environmental impacts of different manufacturing processes over the years. Despite the few attempts to describe the environmental benefits of 3DP [57], a standard method used to quantitatively evaluate the environmental impact “greenness” of different 3DP techniques has not been reported and the literature revealed the need for such metrics [13,58,59]. For the first time, we propose an assessment tool, index of Greenness Assessment of Printed Pharmaceuticals (iGAPP), to evaluate the greenness of different 3DP methods and validate the tool using selected 3D printed dosage forms reported in the literature.

## 2. Index of Greenness Assessment of Printed Pharmaceuticals (iGAPP)

In order to assess the printing process of pharmaceuticals, steps leading to the final products should be taken into account. There are three main stages involved in printing dosage forms. First, the preparation of the printer feed requires mixing of the drug(s), polymer(s), and other excipients uniformly to prepare printing material in a suitable form for the specified printer. Printing material in the form of filaments for FDM, pastes or gels for SSE, powder for BJ and SLS, and solutions for SLA is needed. Secondly, the printing material is fed to the printer, and printing progresses under optimized parameters such as temperature, pressure, speed, and UV intensity. Finally, some products—depending on the adopted printing technology—undergo a post-curing step to remove excess solvent or printing materials or improve mechanical properties. Unlike GC and GAC, the principles of green formulation processes are not defined. As a result, GC principles were considered as a guideline for the selection of the assessment criteria for iGAPP [60]. These include waste reduction, the use of safer solvents, energy efficiency, and the elimination of unnecessary steps. The developed assessment tool, iGAPP, considers the procedures performed in the three steps to provide an inclusive indication of the greenness profile of the whole process. iGAPP indicates the relative greenness of a procedure as opposed to an absolute greenness value. Sometimes it is inevitable to use a particular chemical or follow a high energy-consuming procedure. Nonetheless, the proposed tool provides an opportunity to assess the formulation development procedure and attempt to modify aspects that could contribute to greener practices especially upon upscaling.

### 2.1. Description of iGAPP Tool

The iGAPP illustrates the greenness profile in the form of a pictogram divided into three main sections representing each of the 3DP stages: pre-processing parameters, printing process, and post-curing, and a fourth section in the center presenting an overall score (10 points) of the process with a corresponding shade. Each section is divided into a number of subsections demonstrating individual steps within each stage. Table 2 shows all the subsections of the pictogram and the scores assigned for each step. Steps graded 0, 50, or 100% of the assigned score are shaded green, yellow, or red for green, acceptable, and nongreen procedures, respectively. After evaluating the whole process, the scores for each subsection are summed up, and a total score out of 10 is obtained and included in the center of the pictogram. Total scores higher than 7 are regarded as excellent green procedures, and the middle section of the pictogram is shaded green. 3DP operations with scores 5–7.5 and 0–4.75 are shaded yellow and red and regarded as intermediate green and nongreen procedures, respectively.

#### 2.1.1. Solvent Environmental Impact

Feed preparation is an important step for 3DP of pharmaceutical products. Preparation methods vary with the printing techniques. Solvents constitute a major part of some feeds used for 3DP. For instance, in the case of FDM, the components could be melted together [62] or dissolved in solvent(s) which is then evaporated [63], and the uniform blend is then extruded into filaments. SSE involves the preparation of feed in a gel form and adjusting it to a suitable viscosity [64].

Solvents used in the formulation were considered in the greenness assessment. Judging the safety of solvents has been adopted by several pharmaceutical companies, and solvent selection guides were released. For example, Pfizer constructed a guideline based on worker safety, process safety and environmental and regulatory considerations. The solvents were classified into three categories: preferrable, usable and undesirable. Greener alternatives for undesirable solvents were also included [65]. Similarly, Sanofi devised their guidelines classifying solvents into four classes: recommended, substitution advisable, substitution requested and banned [66]. Although these guidelines give a good overview of the hazardous nature of solvents, it was difficult to compare between solvents within the same category; also, many solvents were not included. Other companies developed guidelines which considered more aspects and provided quantitative evaluation for each solvent. AstraZeneca, GlaxoSmithKline (GSK) and Green Chemistry Institute Pharmaceutical Roundtable created extensive guidelines which displayed health, safety and environmental impacts including life cycle analysis. The solvents were given scores between 1 and 10 accompanied with a corresponding color code (red, yellow, and green). A study was performed comparing 51 solvents using the three guidelines, and it was found that generally there was an agreement between the results of the different guidelines, but they were not identical [10]. Because each guideline based its evaluation on a different number of subcategories, it was challenging to select one of them for iGAPP tool. A decision was made to select GSK because it expanded its guidelines to include 63 more solvents to their initial report covering more possibilities [61]. GSK solvent selection guide for medicinal chemistry ranks solvents according to waste, environmental impact, health, flammability and explosion, reactivity and life cycle. The environmental impact criterion was chosen to grade solvents in the iGAPP because it represents the focus of the tool. Accordingly, each solvent was given a score, represented by 10 points. Solvents that scored ≥ 8 were considered green with low environmental impact and scores ≤ 3 were nongreen solvents inflicting significant harm to the environment where a substitute is encouraged. Solvents with intermediate impact on the environment ranged between 3 and 8. Consequently, the greenness of the solvents used in the pre-processing stage is scored as shown in Table 2.

#### 2.1.2. Temperature Applied in the Feed Preparation Stage

As previously mentioned, some preparatory steps involve a heating element for melting or dissolving the components of the printer feed. In some processes, mixing of the drug and formulation excipients is performed under ambient conditions with no heat, thus conserving energy and satisfying GC concepts [67]. On the other hand, melting requires a higher energy input which varies according to the melting point of the used materials. On that account, procedures carried out at high temperatures are scored lower.

#### 2.1.3. Solvent Removal

Procedures where solvent removal is required before printing pose an environmental threat. Although these solvents are removed from the final product, their utilization during the formulation process exposes personnel to a possible hazard. Additionally, the application of heat to facilitate evaporation is another source of energy consumption.

#### 2.1.4. Number of Active Constituents

Many treatments include more than one drug for synergistic effects and improved efficacy such as cancer therapy [68], relieving different symptoms as in the case of common colds [69], or treating complex conditions such as cardiovascular diseases [70]. Combining more than one active constituent in one formulation saves processing time and excipients needed for the production of individual dosage forms. Moreover, energy consumption to run the manufacturing process will be reduced, as well as utilized packaging materials, therefore regarded as greener.

#### 2.1.5. Energy Consumed for 3DP

Energy consumption is a major concern when it comes to environmental preservation because it is the main source of greenhouse gas emission as most energy demands are met from nonrenewable sources such as oil, coal, and gas [71]. Assuming nonrenewable energy is used, this part of the iGAPP tool is given a higher weight regarding scoring points (2 points). Processes with reduced energy uptake are believed to be more eco-friendly. FDM, SLS, and SLA are the most energy-consuming processes compared to SSE and BJ, requiring heat, high power laser beam, and UV laser beam, respectively. The total energy indicator for most FDM printers was found to be notably greater than SLA and SLS which showed values within close range [27]. A study was reported for modeling of energy consumption of BJ depending on layer thickness [72], whereas limited literature was found on SSE. SSE printers depend on compressors to provide sufficient pressure to extrude the prepared paste, and these compressors consume energy accordingly. More viscous feeds will require higher pressure, so more energy is needed. Therefore, SSE level of energy consumption could vary depending on the printing conditions. Because insufficient studies were available, the used pressure was categorized based on pressure limits available in commercial SSE printers. Some of the most popular SSE-based commercial printers such as BIO X™, Allevi and R-GEN 200 specifications were investigated, and their extrusion pressure ranged from 1 to 980 kPa. The applied pressure during SSE printing was then classified into three ranges (<100 kPa, 100–500 kPa and >500 kPa) and considered excellent, intermediate and nongreen methods, respectively. For the sake of the iGAPP tool, SSE with low extrusion pressure (<100 kPa) and BJ were considered the lowest demanding energy operations because they are carried out under ambient conditions, followed by SLA and SLS then FDM and SSE carried out at extrusion pressure >500 kPa as the highest energy-consuming printers as a result of significant heating and compression energy.

#### 2.1.6. Printing Temperature

Printing temperature varies from one dosage form to the other depending on the printer feed; some require higher temperatures than others. Because heating translates to higher energy usage, printing processes that required no heating are given a green shade and the highest score. According to a review on 3DP of thermolabile drugs, printing temperatures up to 110 °C were considered moderate conditions [73]. Accordingly, temperatures above 110 °C were given red shade and scored 0 points.

#### 2.1.7. Printing Time

A good balance between the printing time and the quality of the finished product is desirable. Faster printing increases the throughput and the efficiency of the 3DP process.

#### 2.1.8. Waste Treatment

Waste from any industrial process presents a serious environmental issue. High amounts of waste reflect on poor production, yield which means excessive consumption of starting materials. Most importantly, disposal is a major concern. One of the advantages of 3DP is the efficiency in utilizing most of the starting materials [74]. However, some printing processes involve the printing of extra structures that act as supports for the created model often performed when using BJ, SLA, and SLS printers [75]. Moreover, extra un-sintered polymers are left at the end of SLS printing. There are three ways to deal with waste: landfill, incineration, or recycling [27]. Ideally, no waste is the most eco-friendly scenario. Alternatively, recycling the waste and reusing it is a valid option that contributes to sustainability. Where recycling is not possible, disposal is inevitable and is scored lowest in the iGAPP tool.

#### 2.1.9. Post Curing Process

Some printing procedures involve post-curing processes for achieving suitable mechanical strength or removal of excess solvent. This additional step could be exposed to ultraviolet light, heating, washing, or a combination of processes [76,77,78]. The post-curing step consumes solvents such as water or alcohol or/and energy in the case of light and heating. Hence, printing techniques that involve no post-curing or non-energy consuming such as drying at room temperature are greener than the ones that do. Simple post-curing that involves heating at low to moderate temperature or washing is accounted as an intermediate green practice, whereas curing with UV light or high temperature is regarded as the least eco-friendly procedure.

#### 2.1.10. Time of Post-Curing Process

The time required for post-curing plays an important factor in energy consumption for drying and UV curing and solvent consumption in the case of washing. For example, drying at 50 °C for 24 h possibly consumes more energy than UV exposure for a minute. Therefore, long post-curing processes (>1 h) are scored less than faster procedures (<1 h)

### 2.2. Validation of the iGAPP Tool

3D printed tablets fabricated by different 3D printing techniques—namely BJ [79], FDM [80], SLA [81], SLS [82], and SSE [83]—were used to validate the developed tool, iGAPP. Greenness assessment pictograms were constructed for each technique and demonstrated in Table 3. Because no information was provided about the waste treatment for the five methods, the following assumptions were assumed: waste from BJ and SLS printers was recycled and from SLA was disposed [27], whereas FDM and SSE did not generate waste because the printed product consumes the entire print feed. Based on the scores, BJ and SLA were found to be the most ecofriendly procedures (7.25 points). BJ consumed minimal energy during pre-processing stage, and printing and water (Environment impact score 10) was used, thus eliminating environmental hazards. The main environmental burdens were the post-curing processes which involved both drying for about 12 h and washing. Similarly, the SLA method was considered green (7.25 points). Whilst it involved waste disposal and consumed more energy than BJ, other green practices such as non-energy consuming post-curing process (washing) and incorporation of four drugs in a single tablet contributed to its greenness. FDM, SLS, and SSE methods were considered intermediate green methods, scoring 5, 5.75, and 5.75, respectively. The absence of solvent or the use of a greener solvent was reflected in the iGAPP tool. FDM and SLS scored higher than SSE in the solvent environmental impact subsection because the FDM method used no solvents and SLS utilized ethanol, which has an environmental impact score of 8 (ranked as a green solvent), whereas propanol used in the SSE method has a score of 7. The methods had other green aspects, namely the lack of waste and post-curing for the FDM method, printing at room temperature, and the possibility of recycling waste for the SLS method and printing at room temperature, and lower energy consuming printer for SSE. However, the FDM method required extrusion and printing at high temperatures, SLS required considerable temperature during printing and solvent removal, and the SSE method involved the use of propanol and the need for a post-curing step.

To evaluate the sensitivity of the tool towards small adjustments, 3DP techniques that employed different feed preparation methods were evaluated. FDM 1 [84] used hot-melt extrusion compared to FDM 2 [85] which applied the solvent casting method. Because FDM 2 utilized methylene chloride (environmnetal impact score 6) and required an extra energy-consuming step for solvent removal, it was scored lower than FDM 1 and regarded as a nongreen procedure (Table 4). Additionally, another comparison between 2 SSE procedures for the fabrication of drug-eluting films was carried out. Both SSE methods were considered to be following green practices regarding solvent environmental impact because SSE 1 [86] did not include solvents and SSE 2f used water for the preparation of the print feed. However, the printing temperature of SSE 1 was higher than SSE 2 where printing occurred at room temperature. Accordingly, iGAPP evaluation scored the latter method higher and was considered a greener method in contrast to SSE 1 (Table 5). Additionally, the extrusion pressure for SSE 1 (600 kPa) was much higher than SSE 2 (65 kPa) and thus scored lower due to higher energy consumption. The iGAPP tool demonstrated acceptable sensitivity and could be used to improve the greenness of a developed fabrication method.

## 3. Limitations and Future Directions

The iGAPP tool provides a simple way of presenting the greenness of 3DP of pharmaceutical formulations, quantitively as well as qualitatively. Nonetheless, there are some limitations; for example, although the impact of hazardous solvents is included in the assessment, the amounts of the solvent are not considered. Moreover, the energy consumption for each printer was assumed to be the same regardless of the dimensions of the printed product, printing bigger products would require higher energy input. Additionally, different models of printers have different energy indicators. This tool was based on small-scale 3DP, so the scoring rubric may differ upon application on a large scale. Further investigation of energy consumption is needed to include printer heating time and consumption as a function of the printed product dimensions, as well as compression power in SSE. Moreover, the inclusion of the amount of generated waste besides treatment could improve evaluation results. Furthermore, the selection of a suitable guide to determine the greenness of solvents was challenging because no standardized guide is available. The available references rank solvents in different ways, so there is a need for unified selection criteria for green solvent selection. This tool provides an indication of the overall greenness of the process and could be used in the optimization phase of the formulation. The presented validation is not universal due to the lack of specific guidelines because the tool is in its preliminary phase.

## 4. Conclusions

The growing interest in 3DP in pharmaceuticals has been evident. As a progressing field, the study of the effect of the process on the environment is necessary. Although some concerns have been discussed in the literature, including energy consumption, waste generation and disposal, and toxicity, no tool for assessing its influence on the environment was created. Hence, a greenness assessment tool for the eco-evaluation of 3DP of pharmaceuticals was developed. The quantitative analysis of the greenness of the applied method coupled with the visual illustration provided by iGAPP offers quick and simple means to compare different methods and define the points of weakness. The tool has also shown accuracy in assessing small changes in methodologies which could help in modifying the method to attempt more sustainable practices.

## Figures and Tables

**Figure 1 pharmaceutics-14-00933-f001:**
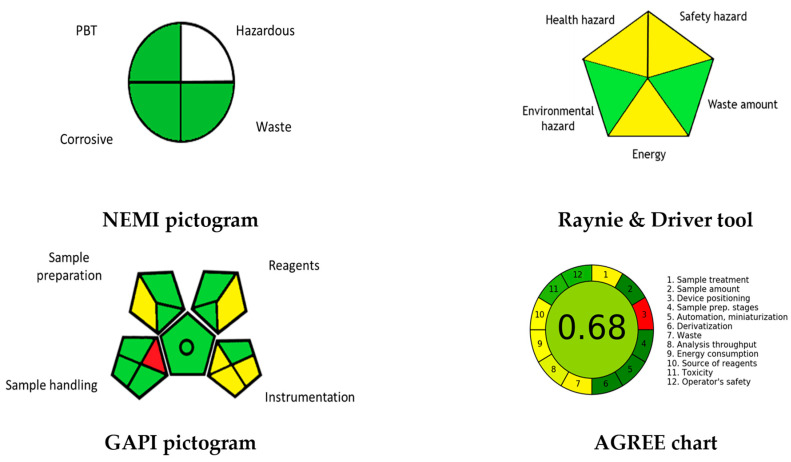
Greenness assessment of a reported HPLC method for the determination of a binary mixture using NEMI, Raynie and Driver, GAPI, and AGREE tools (Adapted from [17]).

**Figure 2 pharmaceutics-14-00933-f002:**
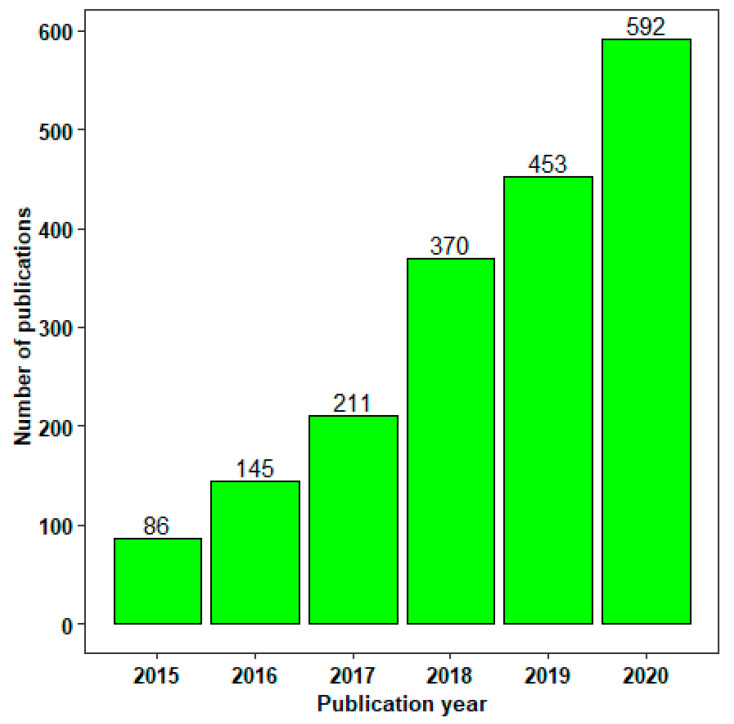
Number of publications involving 3DP techniques by year (2015–2020).

**Figure 3 pharmaceutics-14-00933-f003:**
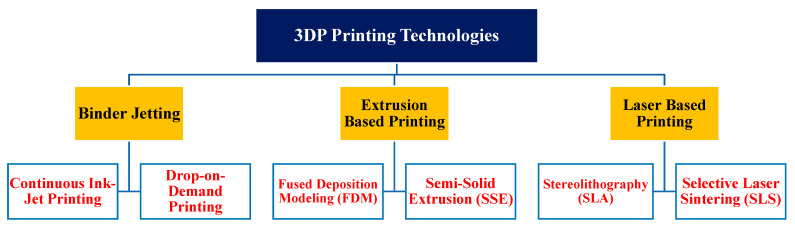
Classification of 3D printing methods.

**Figure 5 pharmaceutics-14-00933-f005:**
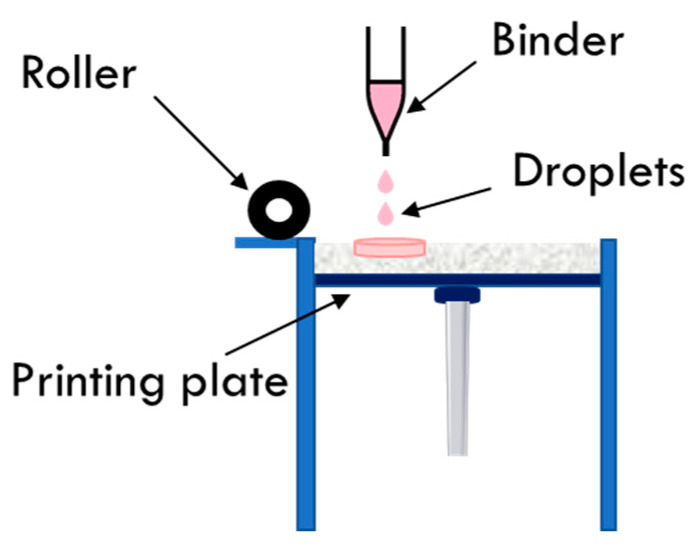
A schematic diagram illustrating BJ printer.

**Table 1 pharmaceutics-14-00933-t001:** The 12 principles of green chemistry summarized by the acronym “PRODUCTIVELY”.

P	Prevent waste
R	Renewable material
O	Omit derivatization steps
D	Degradable chemical products
U	Use safe synthetic methods
C	Catalytic reagents
T	Temperature, pressure ambient
I	In-process monitoring
V	Very few auxiliary substances
E	E-factor, maximise feed in product
L	Low toxicity of chemical products
Y	Yes, it is safe

**Table 2 pharmaceutics-14-00933-t002:** Construction of iGAPP pictogram.

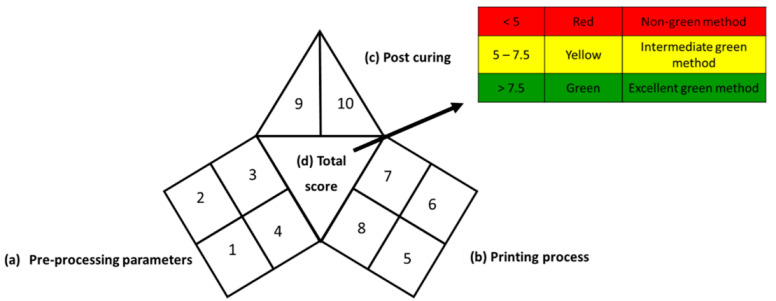				
		**Green** **(Score = 100%)**	**Yellow** **(Score = 50%)**	**Red** **(Score = 0%)**
(a) Pre-processing parameters				
1	Solvent environmental impact * (1 point)	Environmental impact score: ≥8	Environmental impact score: 4–7	Environmental impact score: ≤3
2	Temperature (°C) (1 point)	<30	30–60	>60
3	Solvent removal (1 point)	No solvent removal required	Evaporation at room temperature	Evaporation at temperature >25
4	No. active constituents (1 point)	>2	2	1
(b) Printing process				
5	Energy consumption (2 points)	BJ SSE (<100 kPa)	SLA SLS SSE (100–500 kPa)	FDM SSE (>500 kPa)
6	Temperature (°C) (1 point)	Room temperature	26–10	>110
7	Printing time per product (min) (1 point)	<2.5	2.5–10	>10
8	Waste treatment (1 point)	No waste	Waste is recycled	Waste is disposed
(c) Post curing				
9	Post curing process (0.5 point)	No post-curing/ non-energy consuming process	Drying at temperature <60 °C	Higher energy-consuming post-curing process
10	Time of post-curing process (hours) (0.5 point)	No post-curing	<1	>1
(d) Total score				
		>7.5	5–7.5	<5

* GlaxoSmithKline solvent selection guide [61].

**Table 3 pharmaceutics-14-00933-t003:** Greenness profile of 3DP tablets by reported BJ, FDM, SLA, SLS, and SSE methods.

	BJ [79]	FDM [80]	SLA [81]	SLS [82]	SSE [83]
iGAPP pictogram	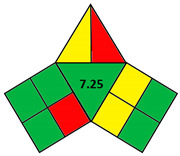	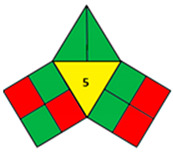	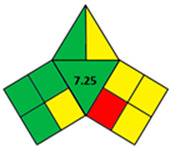	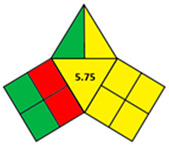	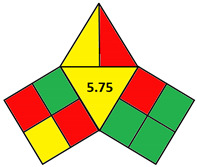
Solvent environmental impact	Water: 10 (1)	No solvent (1)	No solvent (1)	Ethanol: 8 (1)	Propanol: 7 (0.5)
Temperature (°C)	Room temperature (1)	80–110 (0)	Room temperature (1)	Room temperature (1)	70 (0)
Solvent removal	No solvent removal (1)	No solvent removal (1)	No solvent removal (1)	40 °C (0)	No solvent removal (1)
No. active constituents	1 (0)	1 (0)	4 (1)	1 (0)	1 (0)
Energy consumption	BJ (2)	FDM (0)	SLA (1)	SLS (1)	SSE (55–65 kPa) (2)
Temperature (°C)	Room temperature (1)	180–190 (0)	Room temperature (1)	80–100 (0.5)	Room temperature (1)
Printing time per product (min)	2.5 (0.5)	<2.5 min (1)	2.5–10 (0.5)	<2.5 min (1)	11–14 min (0)
Waste treatment	Recycle (0.5)	No waste (1)	Waste is disposed (0)	Recycle (0.5)	No waste (1)
Post curing	Drying 40 °C (0.25)	No post curing (0.5)	non-energy consuming process (0.5)	non-energy consuming process (0.5)	Drying at room temperature (0.25)
Time of post-curing process (hours)	Washing + Drying (>1 h) (0)	No post curing (0.5)	Washing (<1 h) (0.25)	Powder removal (<1 h) (0.25)	Drying (>1 h) (0)

**Table 4 pharmaceutics-14-00933-t004:** Comparative greenness evaluation of two reported FDM methods.

	FDM 1 [84]	FDM 2 [85]
iGAPP pictogram	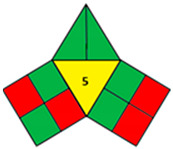	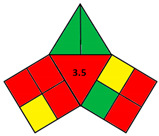
Solvent environmental impact	No solvent (1)	Environmental impact score: 6 (0.5)
Temperature (°C)	170–200 (0)	140 (0)
Solvent removal	No solvent removal (1)	Solvent removal (0.5)
No. active constituents	1 (0)	1 (0)
Energy consumption	FDM (0)	FDM (0)
Temperature (°C)	195–208 (0)	164 (0)
Printing time per product (min)	<2.5 min (1)	2.5–10 (0.5)
Waste treatment	No waste (1)	No waste (1)
Post curing	No post-curing (0.5)	No post-curing (0.5)
Time of post-curing process (hours)	No post-curing (0.5)	No post-curing (0.5)

**Table 5 pharmaceutics-14-00933-t005:** Comparative greenness evaluation of two reported SSE methods.

	SSE 1 [86]	SSE 2 [87]
iGAPP pictogram	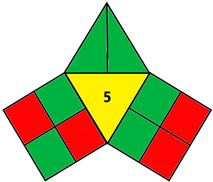	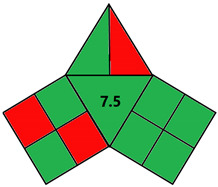
Solvent environmental impact	No solvent (1)	Environmental impact score: 10 (1)
Temperature (°C)	140 (0)	70 (0)
Solvent removal	No solvent removal (1)	No solvent removal (1)
No. active constituents	1 (0)	1 (0)
Energy consumption	SSE (600 kPa) (0)	SSE (65 kPa) (2)
Temperature (°C)	140 (0)	Room temperature (1)
Printing time per product (min)	<2.5 min (1)	<2.5 min (1)
Waste treatment	No waste (1)	No waste (1)
Post curing	No post-curing (0.5)	Drying at room temperature (0.5)
Time of post-curing process (hours)	No post-curing (0.5)	Drying (>1 h) (0)

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
