# Peer review of "Development and Validation of a Novel Tool for Assessing the Environmental Impact of 3D Printing Technologies: A Pharmaceutical Perspective"

_pharmaceutics, 2022, doi:10.3390/pharmaceutics14050933_

Round 1

Reviewer 1 Report

A very welcomed article making a great attempt to create a quantitative tool, i.e. the index of Greenness Assessment of Printed Pharmaceuticals, that evaluates the greenness of the different 3DP technologies used in the pharmaceutical industry. The paper is well structured proceeding from a general introduction to covering the general aspects of different printing techniques to the actual description of the Tool and examples of its use.

It is clear that this is first attempt to do this and it the classifications of the points in the different areas are somewhat arbitary and not very accurately defined. It would be advisable to add a few statements that this tool and the points given still needs to be further developed to be more accurate and it not be used as such without critical assessment of each factor used. For instance the post curing step should be evaluated based on real values of energy consumption not by only stating if there is curing step with UV (eg does it take 5 s or 1 min makes a big difference). Also it might be useful to look into the ranking within each a category  (i.e. printing time might be much lower than indicated in any of the examples seconds rather that minutes, what would the scoring then be?) So please elaborate further on the limitations and better to state that the validation presented is of course not very universal as the tool is so young and still under-developed.

In any case, a very nice paper and welcomed contribution in the field!

Reviewer 2 Report

The current manuscript provides and interesting account of ascertaining the greenness of 3D printing technologies. There are some suggestions that I would like to make as follows:

1. Greenness: The title should rather include "environmental impact"?

2. The are some parameters missing from the tables and assessments. For examples, the preprocessing of samples before printing (especially for hydrogels). Also for the pressure system used in nozzle based bioink printing, the energy consumption and space used for the compressor needs to be accounted into. In the same line, the use of physical space and drying of samples post-fabrication need to be considered.
